# A Systematic Review of Individualized Heart Surgery with a Personalized Prosthesis

**DOI:** 10.3390/jpm13101483

**Published:** 2023-10-11

**Authors:** Faizus Sazzad, Kollengode Ramanathan, Irwan Shah Moideen, Abdulrahman El Gohary, John Carey Stevens, Theo Kofidis

**Affiliations:** 1Department of Surgery, Yong Loo Lin School of Medicine, National University of Singapore, Singapore 117599, Singapore; 2Department of Cardiac, Thoracic and Vascular Surgery, National University Heart Centre, Singapore 119228, Singapore

**Keywords:** personalized, individualized, open-heart surgery, implantation, systematic review

## Abstract

Personalized surgery is not just a new trend but rather a patient-specific approach to therapy that makes it possible to adopt a targeted approach for a specific patient and closely mirrors the approach of personalized medicine. However, the application of tailored surgery in the context of cardiovascular replacement surgery has not been systematically reviewed. The ability to customize a device is highly dependent on the collection of radiological image data for precise prosthesis modeling. These facts are essential to “tailor-made” device design for precise prosthesis implantation. According to this study, computed tomography (CT) was the most prominent imaging modality; however, transesophageal echocardiography and echocardiography were also found to be helpful. Additionally, a dynamic finite element simulation was also found to be an attractive alternative to the finite element analysis for an in-silico experiment. Nonetheless, there is a paucity of relevant publications and only sporadic evidence. More clinical studies have been warranted, notwithstanding that the derived data and results from this insight into the use of therapeutic interventions may be evidence of multiple directives in clinical practices and beyond. This study may help the integration of personalized devices for better comprehension of predicted clinical outcomes, thus leading towards enhanced performance gains.

## 1. Introduction

Precision medicine, with a specific focus on personalized surgery, has been an evolving field of practice since the late 1990s [1]. Over the past decade, significant progress has been made, largely attributed to advancements in technology, notably the cost reduction of three-dimensional computational modeling. These technological breakthroughs have ignited growing interest and applications in various surgical subspecialties [2]. Personalized surgery has gained immense importance across a wide spectrum of surgical disciplines by facilitating comprehensive preoperative assessments of individual patients. This process involves the meticulous consideration of predefined variables, allowing surgical procedures to be tailored precisely to the unique needs and characteristics of each patient [3,4,5]. This transformative shift in the approach to surgery has had a profound impact on the field of cardiovascular surgery, extending its reach and capabilities. By embracing the principles of personalized surgery, cardiovascular surgeons can now offer highly customized treatment strategies that account for the specific anatomical, physiological, and genetic traits of their patients. This not only enhances the precision and effectiveness of surgical interventions but also contributes to improved patient outcomes and overall healthcare quality. 

Personalized cardiac surgery represents a meticulous approach in which surgical techniques are tailored to suit the specific needs and attributes of an individual patient’s heart [6]. This method acknowledges the inherent uniqueness in the anatomy, functionality, and pathological conditions of each patient’s heart. Consequently, relying on a standardized or universally applicable surgical approach may not lead to the most favorable outcomes. Interestingly, it is important to note that the application of individualized surgical methods in the context of cardiovascular replacement surgery has not been comprehensively examined or assessed in previous research endeavors. Rather than adhering to a one-size-fits-all or ready-made surgical strategy, this approach underscores the importance of tailoring surgical techniques to optimize results and address the specific needs of the patient’s heart. Remarkably, despite the potential benefits, the systematic examination of personalized surgical techniques in the realm of cardiovascular replacement surgery has been somewhat lacking in prior research.

Traditionally, cardiovascular implant surgery has predominantly focused on replacement procedures aimed at addressing diseased heart valves [7,8,9]. While aortic and mitral valve diseases are the most prevalent and severe, necessitating invasive interventions, valve-related issues can affect any single valve or a combination of the four [10,11,12,13]. Treatment modalities for heart valve disease encompass various approaches, ranging from invasive methods such as tissue or mechanical heart valve replacement and repair products to less invasive techniques like percutaneous balloon valvuloplasty and minimally invasive transcatheter procedures [14,15,16]. However, the current prosthetic solutions exhibit limitations. They often involve the insertion of a rigid, circular implant at the level of the patient’s native valve annulus, which may not fully account for the intricacies of individual anatomy. Furthermore, commercially available prostheses, characterized by their inflexible and symmetrical designs, are highly dependent on medications with significant side effects, potentially resulting in serious health complications [17,18,19].

One of the current challenges that warrants attention is the matter of patient–prosthesis mismatch (PPM), which has received significant attention, particularly in the context of heart valve replacement therapy. PPM is a term frequently discussed in clinical circles, referring to a scenario wherein a surgically implanted medical device is, in essence, too small in proportion to the patient’s needs, thus giving rise to a condition reminiscent of “valve stenosis.” This predicament has provoked criticism of surgeons, as it seems that they are effectively replacing one medical ailment with another. The inadequacies inherent in the design of commercially available prosthetic devices only serve to underscore the urgent necessity for a concrete solution, one that can be achieved through the widespread adoption of personalized prosthetic devices.

The primary goal of personalized prostheses is to create a device or implant that is tailored specifically to suit the unique needs of an individual recipient. This approach has found considerable success in various fields, such as orthodontics, orthopedics, and reconstructive surgery. Despite these successes, it is noteworthy that the concept of personalization has not yet been widely embraced in the realm of cardiovascular replacement therapy. As of now, there are no commercially available personalized devices designed for this particular subspecialty, although some are currently undergoing evaluation and development. Given this situation, it becomes imperative to consider and explore the potential benefits and applications of personalized prostheses in the context of cardiovascular replacement therapy. In essence, the focus is on creating prosthetic solutions that are not one-size-fits-all but rather precisely customized to meet the distinct requirements and anatomical characteristics of individual patients. While this approach has demonstrated its effectiveness in other medical disciplines, it is an exciting frontier that holds significant promise for enhancing the outcomes and quality of life for individuals in need of cardiovascular replacement therapy.

Furthermore, prosthetic devices come in a few predetermined sizes and are offered off-the-shelf during the surgical procedure itself, based on rough eyeballing or subjective assessments [20,21,22,23]. Similarly, aortic surgery and concomitant cardiac procedures like left atrial appendage occlusion are not addressed in precision but rather largely dominated by the standardized prosthesis and subjective assessment [24,25]. Hence, this review of the application of personalization in cardiovascular replacement surgery will help identify opportunities for expanded surgical adaptation and future research directions that can address the rational challenges in this field.

## 2. Materials and Methods

### 2.1. Search Strategy

A comprehensive search of the existing literature was conducted using the Preferred Reporting Items for Systematic Reviews and Meta-Analyses (PRISMA) guidelines [26]. This search was carried out electronically and encompassed multiple databases, including Medline (via PubMed), Scopus, Embase, and Web of Science, spanning the period from their inception up to 29 July 2023. To ensure a thorough and exhaustive retrieval of pertinent literature, a meticulous search strategy involving a combination of Medical Subject Headings (MeSH) terms was employed. These MeSH terms included “precision medicine”, “personalized”, “individualized”, “heart valve”, “aortic”, “cardiovascular surgical procedures”, “mitral valve replacement”, “mitral valve implantation”, “transcatheter”, “transcutaneous”, and “transcatheter mitral valve replacement”. The published papers extracted as a result of this search underwent a rigorous and systematic screening and assessment process.

### 2.2. Inclusion Criteria and Exclusion Criteria

This analysis encompassed a range of research methodologies, specifically focusing on studies related to the utilization of personalized prosthetic devices in cardiovascular replacement surgery. The eligible types of studies considered in this review comprised randomized controlled trials, prospective observational investigations, interventional studies, experimental research, retrospective cohort studies, and cross-sectional studies. These studies were selected because they provided valuable insights into the application of personalized prostheses in the context of cardiovascular surgery. 

However, certain types of research were deliberately omitted from this analysis. Articles that involved concomitant medical procedures, surgical repairs, cases related to congenital or adult congenital heart disease surgery, survey-based investigations, and studies not written in the English language were excluded from our examination. These exclusions were made with the intention of maintaining the concentration of our analysis on the particular field of individualized prostheses in cardiovascular replacement surgery and omitting irrelevant information or studies that would have presented language hurdles.

### 2.3. Study Selection and Outcome of Interest

Three authors conducted a thorough and systematic review of the available studies independently in order to determine their eligibility for inclusion in their research. The initial screening of published articles involved an examination of their titles and abstracts, with intentionally broad criteria employed to ensure that all potentially relevant studies were considered. To manage the vast array of extracted citations, the authors utilized reference software called EndNote X9 [27]. In the subsequent stage of the review process, studies that had successfully passed the initial screening were subjected to a more in-depth evaluation. 

In cases in which a definitive decision could not be reached during the initial screening, a full-text review of the articles was carried out to confirm their relevance to the research. During this stage, each author independently abstracted key details from the selected studies, including information about the study’s design, applied research methods, the model of personalization utilized, the intended purpose of the study, and the observed outcomes. In addition to the information extracted from the selected articles, the authors conducted a manual search to identify any additional relevant data using a backward snowballing method. This comprehensive approach ensured that the research was based on a thorough examination of all pertinent literature, providing a solid foundation for the study.

### 2.4. Quality of Evidence and Risk of Bias Assessment

The research encompassed a collection of studies, all of which fell into the categories of observational or experimental investigations, with the majority of these studies primarily focusing on the initial outcomes of their respective experiments. To gauge the quality of evidence within these studies, the GradePro tool was employed. Additionally, in accordance with the guidance provided in Chapter 11 of the *Cochrane Handbook for Systematic Reviews of Interventions*, the ROBINS-I tool (Risk of Bias in Non-randomized Studies-of Interventions) was utilized to assess the potential for bias in studies that lacked randomization [28]. In order to comprehensively evaluate the articles included in this research endeavor, an assessment of their risk of bias and the overall quality of evidence was conducted. This evaluation was facilitated through the use of statistical software known as Revmen 5.4 [29]. To ascertain the risk of bias in the studies that formed part of this investigation, the assessment was carried out in accordance with the guidelines set forth in Chapter 8 of the *Cochrane Handbook for Systematic Reviews of Interventions* [28].

## 3. Results

A comprehensive search across multiple databases, including Medline (via PubMed), Scopus, Embase, and Web of Science, yielded a total of 514 potential articles. These articles were identified electronically as potential sources of information. After rigorous screening based on predefined inclusion criteria, seven studies were chosen for further analysis in order to synthesize evidence. These criteria were established in advance to ensure the selection of relevant and high-quality studies. In these seven studies, a total of 229 individuals who had received personalized prostheses for cardiovascular replacement therapy were included as participants. For a visual representation of this selection process, please refer to Figure 1.

### 3.1. Quantity of Evidence

The preliminary systematic scrutiny of our study involved a meticulous and systematic examination of electronic sources, which yielded a total of 512 published articles. To ensure comprehensiveness, we conducted an additional, iterative search on platforms such as Google Scholar, ResearchGate, and the Ubiquity Partner Network, which led us to discover two more articles of relevance. Subsequently, employing citation management software, specifically Endnote X9 [27], we meticulously removed duplicate entries, resulting in a refined collection of 221 published articles for further scrutiny and evaluation.

To efficiently narrow down our selection, we initially scrutinized the titles and abstracts of these 221 papers using Endnote X9. Articles that did not align with our predefined inclusion and exclusion criteria were excluded from the subsequent stages of our review process. As a result, we retained 22 articles for in-depth examination, while 199 records were excluded from further consideration. Moving forward, a comprehensive assessment of the 22 selected articles was conducted by performing a full-text review. This phase of our evaluation process led to the exclusion of studies that did not fit the specific focus areas of our research, including those centered on congenital heart surgery (n = 7) or immunotherapy after heart transplantation (n = 5), as well as review articles (n = 2) and editorial/commentary pieces (n = 1). Consequently, a final set of seven articles, spanning references [30,31,32,33,34,35,36], were deemed suitable for evidence synthesis and, ultimately, were included in our study for analysis (as detailed in Appendix A). 

### 3.2. Included Studies Characteristics

The seven studies encompassed in this analysis [30,31,32,33,34,35,36] were conducted at single medical centers, with the majority of them being situated in the United States and various European countries. One notable exception was a study carried out in China [36], as indicated in Table 1. These studies exhibited significant heterogeneity in both the choice of medical procedures employed and the selection criteria for study participants. It is worth noting that the majority of personalized approaches were focused on human subjects, although two studies, namely Amerini et al. [30] and Robinson et al. [35], employed experimental large animals for the purposes of testing and evaluating medical devices.

Among the studies included in this review, only one was a clinical trial conducted by Collis et al. [31], which assessed personalized surgical strategies rather than the assessment of a newly designed prosthetic device. It is noteworthy that computed tomography imaging was the prevailing modality employed for preoperative assessments in most of the studies included in this review. However, some investigators opted for alternative modalities, such as echocardiography [31], transesophageal echocardiography [34], and 3D transesophageal echocardiography [35] for their evaluations.

### 3.3. Summary of the Intervention

A comprehensive overview of interventions conducted across all the studies included in our analysis can be found in Table 1. In the pursuit of crafting tailored prosthetic solutions, the majority of the cases opted for an “in silico” approach, relying on Finite Element Analysis. It is noteworthy that Collis et al. [31] deviated from this trend by employing a clinical assessment methodology specifically based on preoperative echocardiography. The utilization of software tools for the prototyping process exhibited a degree of diversity among the studies we reviewed. However, a common thread across these investigations was the utilization of computer-aided design in conjunction with 3D rendering techniques for the reconstruction of anatomical models. Notably, there were variations observed in the final design of personalized prosthetic devices in the studies, reflecting a degree of customization to cater to individual patient needs. There was variation in the final model or the personalized devices under study, but most of the prostheses were modeled by three-dimensional printing using various artificial printing materials. None of the studies used 3D bioprinting materials.

### 3.4. Outcomes and Interest Variables

The data summarized in Table 1, which has been gathered from seven different research studies involving a total of 229 recipients, reveal a consistent trend of utilizing either surgical procedures or catheter-based interventions, with one exception in the study by Rim et al. [34], which employed a virtual dynamic finite element simulation. Following these interventions, the majority of the devices underwent assessment through CT scans and echocardiography, encompassing both 2D and 3D imaging techniques. However, it is important to note that Rim et al. [34] relied on finite element analysis (FEA), while Yuan et al. [36] utilized angiographic evaluation instead. Notably, Amerini et al. [30] and Robinson et al. [35] took a different approach by testing the devices in large animals, specifically porcine and canine subjects, respectively. Subsequently, these studies conducted postoperative autopsies after a predetermined period in accordance with their respective study designs.

## 4. Discussion

This comprehensive systematic review explores the field of personalized therapy in cardiovascular replacement surgery and highlights the idea of personalizing and the concept of tailoring surgical procedures to meet the unique needs of each patient. It offers a meticulous synthesis of evidence supporting the notion of “personalized surgery” and its pivotal role within the cardiovascular surgery domain. The review not only explores the significance of personalization but also delves into the intricacies of the personalization process and the various interventional methods employed and offers a qualitative assessment of its efficacy. Presently, the landscape of cardiovascular intervention devices available in the commercial market remains devoid of individualization [25,37]. Specifically, the vast majority of prosthetic heart valves are offered in a limited range of three or fewer predefined sizes, further compounded by the constraints imposed by the hospital’s inventory management system. This predicament leaves cardiovascular surgeons with only premanufactured options when embarking on the implantation procedure. Consequently, surgeons are left with no choice but to utilize these rigid, circular designs for insertion into a human heart, all the while harboring unrealistic hopes for long-term success. This challenge is further exacerbated by concerns related to PPM.

Although there is an obvious absence of specialized equipment created especially for tailored surgical operations, it is important to note that researchers and medical professionals have acknowledged the relevance and necessity of filling this gap. Furthermore, it is noteworthy that a broad range of target regions have adopted individualized methods and prostheses in cardiac surgery. Many researchers have explored various techniques and treatment plans in this area. An important finding from the literature at hand is the wide range of preoperative imaging techniques used to collect the crucial recipient information needed to develop customized surgical models. These imaging modalities cover a variety of procedures, such as computed tomography scans, magnetic resonance imaging, preoperative echocardiography, and three-dimensional transesophageal echocardiography. These instruments frequently provide comprehensive anatomical information to assist in surgical planning. The lack of a unified opinion on the “optimal imaging modality” for the construction of three-dimensional representations and the collecting of preoperative scan data is also an important topic to underline. This lack of agreement emphasizes the continual investigation and assessment of diverse imaging techniques in the pursuit of accuracy and efficacy in individualized surgical operations.

However, various researchers used diverse diagnostic imaging techniques in their individual studies. The principal imaging technique used by Amerini et al. [30], Pasta et al. [33], and Yuan et al. [36] was CT scanning. In contrast, Collis et al. [31] showed a preference for echocardiography. Transesophageal echocardiography was successfully used by Rim et al. [34], but it was interesting to see that Robinson et al. [35] took it a step further by including 3D transesophageal echocardiography in their strategy. Before any planned procedure, each patient underwent at least one echocardiography scan, an essential step in gathering preoperative data for personalized care. Additionally, some patients also underwent preoperative CT scans as a routine practice, which proved valuable in tailoring imaging to collect individualized data. It is worth highlighting that, over the past decade, there has been a remarkable improvement in the quality of CT scans, making them more patient-centric [38]. More individualized and accurate diagnostics are becoming more common in the world of transcatheter valves. Multidetector computed tomography (MDCT) scans and sophisticated 3D reconstructions are used to take exact measurements. This method improves decision-making, especially when choosing the right prosthesis type and size for each patient’s particular requirements. Yet, that does not impact manufacture and prosthesis design. A combination of both modalities could be a better approach to generating a larger preoperative data pool. 

In the field of current practice, it is customary to employ the data obtained from preoperative imaging of patients for this purpose, followed by subsequent “in silico” processing using finite element analysis and/or three-dimensional personalized modeling techniques [39,40,41]. The utilization of three-dimensional computer-aided design (3D CAD) and finite element analysis has proven to be highly effective in creating in silico models. These models are constructed with the assistance of advanced geometry and mesh generation software, including but not limited to SolidWorks, Rhino CAD, and MIMIC 16.0 [33,35,36]. In the majority of studies reviewed, a consistent methodology was employed, commencing with the development of a virtual 3D model. This model was then subjected to a 3D rendering process and, subsequently, personalized 3D printing [39,40]. It is worth noting, though, that the specific form and functionality of the resultant devices varied significantly, contingent on the targeted organ or the therapeutic objectives of the procedure. 

The personalized prostheses generated in these studies were crafted from materials such as artificial alumide mold, nitinol, and silicone [30,32,35]. Notably, none of these devices were created using 3D bioprinting techniques. This exclusion can be attributed to the inherent limitations and challenges associated with 3D bioprinting in the context of cardiac tissues [42,43]. Postoperative verification of the precision of the surgical prostheses was routinely carried out, primarily relying on computed tomography (CT) scans and echocardiography as the standard modalities for these assessments. However, a promising avenue emerged in the form of virtual dynamic finite element simulations, which demonstrated their potential for accurately estimating and quantifying the motion of mitral valve leaflets [34,44]. It is interesting to note that, in the cases we evaluated, individualized prostheses were positioned consistently for both surgical and transcatheter procedures. This finding is consistent with the growing preference for catheter-based therapies, indicating the growing importance of these procedures in the field of cardiovascular replacement therapy.

### 4.1. Limitations 

Firstly, it is essential to emphasize that this systematic review is exclusively centered on research concerning personalized cardiovascular replacement surgery, which represents a highly specialized and focused area of investigation. Secondly, the findings of this study rely on experimental research with preliminary data, which has limitations concerning procedural responses and the reproducibility of trend data. This element imposes some limitations on the replicability of trend data as well as on procedural replies. The reviewed trials reveal notable differences in their emphasis on cardiovascular interventions, likely due to the limited empirical data available in this specialized field, resulting in significant heterogeneity in the specific interventions addressed. A notable limitation of this study is the restricted utilization of large animal models in the conducted experiments, a factor that impacts the statistical significance of the obtained results. Lastly, it becomes evident that there is a pressing need for additional randomized clinical trials and expansive clinical investigations to tackle outstanding questions and offer a more comprehensive assessment of personalized prosthetic applications in actual clinical scenarios.

### 4.2. Future Directions

The scope of future directions in the research in this field is multifaceted and encompasses a range of crucial areas. Firstly, there is a need for a comprehensive systematic review that goes beyond the confines of personalized cardiovascular replacement surgery. This expanded review should encompass the entire spectrum of cardiovascular diseases and the various treatment methods that fall under the umbrella of personalized cardiovascular medicine. Moreover, it is essential to explore and delve into the application of individualization methodologies in diverse surgical subspecialties, including but not limited to orthopedics, neurosurgery, and plastic surgery. By adopting this interdisciplinary approach, we can contribute significantly to gaining a more profound and holistic comprehension of personalized prostheses throughout the medical field.

Another avenue for further investigation involves conducting specific comparative studies focusing on personalized prostheses in distinct cardiovascular replacement surgeries, such as heart valve replacement. Such comparative studies are pivotal as they can provide insights into the advantages and limitations of personalized approaches in various cardiac procedures, thereby shedding light on the relevance and effectiveness of tailoring treatments to individual patients in the domain of cardiovascular care. Lastly, there is a pressing need for rigorous clinical trials and large-scale studies. These studies should be designed to address lingering questions regarding the practical utilization of personalized prostheses in real-world clinical settings. The accumulation of robust clinical evidence is of paramount importance, as it serves to establish the safety and efficacy of personalized approaches in the realm of patient care. This empirical foundation is essential for ensuring that personalized medicine can be confidently integrated into standard healthcare practices, ultimately benefiting patients across the board. 

## 5. Conclusions

The concept of a universal “one-size-fits-all” approach continues to persist in practical application, especially within the cardiovascular domain of what is often considered the “gold standard therapy.” However, it is crucial to acknowledge the limitations of such an approach, as it essentially caters to only a select few, failing to adequately address the needs of many. This limitation becomes even more apparent when we consider that for some individuals, the conventional approach hardly fits at all. This situation is particularly evident in the context of healthcare, given the increasing prevalence of comorbidities and the intricate and diverse ways in which diseases manifest in patients. Nevertheless, there is hope on the horizon. The adoption of personalized prosthetic solutions has the potential to yield significant improvements in patient-specific clinical outcomes.

We envision and believe that cutting-edge technological advancements will be crucial in ushering in a new era of highly individualized healthcare. This change will involve a variety of medical procedures, including tablets, tests, and implants that are customized to each patient’s particular anatomical characteristics and metabolic or biological profile. This marks a shift from the prevalent model of mass-producing generic medical products and places more emphasis on the significance of individualized treatment as the future’s gold standard. 

## Figures and Tables

**Figure 1 jpm-13-01483-f001:**
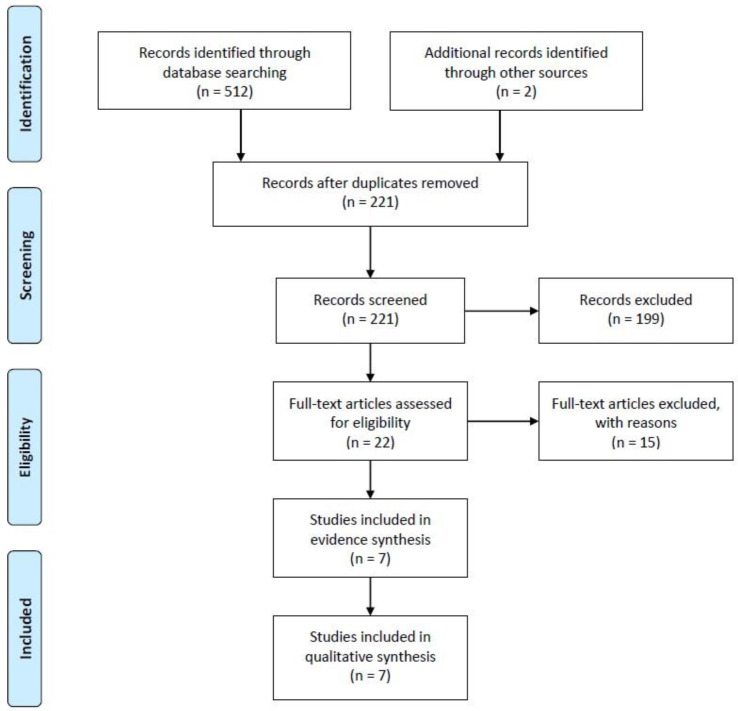
PRISMA flow diagram showing the enrolment of the eligible articles for the systematic review.

**Table 1 jpm-13-01483-t001:** Characteristics of the included studies of personalized cardiovascular replacement surgery.

Author, Year	Amerini et al., 2014 [30]	Collis et al., 2018 [31]	Ovcharenko et al., 2016 [32]	Pasta et al., 2020 [33]	Rim et al., 2015 [34]	Robinson et al., 2018 [35]	Yuan et al., 2017 [36]
Journal	ICVTS	EJCTS	CBM	MBEC	PLOS 1	NBE	SRS
Place	Germany	UK	Russia	Italy	Texas	New York	China
N	12	203	1	9	1	1	2
Species	Pigs	Patients	Patients	Patients	Patients	Canine	Patients
Study type	Experimental	Clinical trial	Case report	Case series	Case report	Experimental	Case reports
Target site	Tricuspid valve	LVOT	Aortic valve	BAV	Mitral valve	LAA	Aorta
Method	CT scan	Echo	CT scan	CT scan	TEE, FEA	3D TEE	CT scan
Prototyping	In silico	Clinical	FEA	Rhino CAD	FEA	SolidWorks CAD	MIMIC 16.0
Postprocessing	RA geometry	HCM	3D modeling	Aortic valve	MVs	LA occlusion	3Dp model
Intervention	Catheter guided	Surgery	TAVR	TAVI	Virtual repair	Guided surgery	Simulation Surgery
Reconstruction	3D	NA	3D	ICEM-CFD	Virtual MV	3D	3D
Model	Solid alumide	Surgical technique	Nitinol model	Cobalt-chromium	Dynamic FEA	Silicone	TADA 3D printed
Device	TTVI	NA	CoreValve	SAPIEN 3	Mitral valve leaflet	LAA occlude	Chimney stents
Positioning	Surgical	Surgical	Transcatheter	Transcatheter	Virtual	Surgical	Transcatheter
Evaluation	CT scan	Echo	CT scan	CT scan and TEE	Dynamic FEA	3D TEE	Angiography
Autopsy	All cases	-	-	-	-	After 24 h	-
Journal	ICVTS	EJCTS	CBM	MBEC	PLOS 1	NBE	SRS
Relevant image	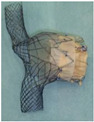	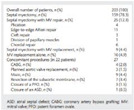	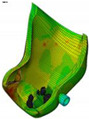	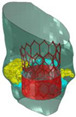	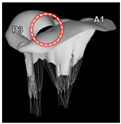	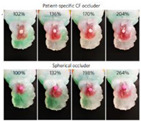	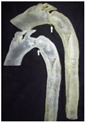

NB. ICVTS: *Interactive Cardiovascular and Thoracic Surgery Journal*; RA: Right atrium; 3D: Three dimensional; TTVI: Transcatheter tricuspid valve implant; CT: Computed tomography; EJCTS: *European Journal of Cardio-Thoracic Surgery*; UK: United Kingdom; LVOT: Left ventricular outflow tract; HCM: Hypertrophic cardiomyopathy; NA: Not applicable; CBM: *Computers in Biology and Medicine*; FEA: Finite element analysis; TAVR: Transcatheter aortic valve replacement; MBEC: *Medical & Biological Engineering & Computing*; BAV: Bicuspid aortic valve; CAD: computer-aided design; TAVI: Transcatheter aortic valve implantation; ICEM-CFD: Software package which provides advanced geometry/mesh generation by Ansys Inc.; TEE: Transesophageal echocardiography; PLOS One: *Public Library of Science One* j; MVs: Degenerative mitral valves; A1: First segment of the anterior mitral leaflet; P3: Third segment of the posterior mitral leaflet; red circle: Highlighted severe mitral valve prolapse; NBE: *Nature Biomedical Engineering*; LAA: Left atrial appendage; LA: Left atrium; CF occlude: Cauliflower occlude; SRS: *Scientific Reports*; 3Dp: Three-dimensional printing. TEE: Transesophageal echocardiography; LAA: Left atrial appendage; TADA: Thoracic aortic dissection aneurysm; white arrows: Indicates the maximum angle of the arch.

## Data Availability

The authors confirm that the data supporting the findings of this study are available within the article and its Appendix A. Derived data supporting the findings of this study are available from the corresponding author on request.

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
