# Peer review of "A Systematic Review of Individualized Heart Surgery with a Personalized Prosthesis"

_jpm, 2023, doi:10.3390/jpm13101483_

Round 1

Reviewer 1 Report

Sazzad and Couleges address the use of individualized prostheses in cardiac surgery in their article "A Systematic Review of Individualized Heart Surgery with a Personalized Prosthesis."

Although the topic is important and will be even more important in the future, the authors did not describe it in detail. They compare different types of studies (case report, experimental setting, animal setting, clinical trial) and different prostheses. They did not summarize and present the different studies properly. All in all, the article is very confusing and not suitable to be published. 

I thank the authors for the opportunity to review the paper.

Please check the English spelling

Author Response

We greatly appreciate your considerate feedback and the invaluable period of time you so kindly spent reading through our work. We truly appreciate your input, which is crucial in assisting us in enhancing and refining our research. We have included a point-by-point response to the comments and questions of the reviewers in the following section:

Reviewer-1

Sazzad and Couleges address the use of individualized prostheses in cardiac surgery in their article "A Systematic Review of Individualized Heart Surgery with a Personalized Prosthesis."

Although the topic is important and will be even more important in the future, the authors did not describe it in detail. They compare different types of studies (case report, experimental setting, animal setting, clinical trial) and different prostheses. They did not summarize and present the different studies properly. All in all, the article is very confusing and not suitable to be published. 

I thank the authors for the opportunity to review the paper.

Response:

We acknowledge your review and comments. In response to your feedback, we have made substantial revisions to the manuscript. We have addressed the concerns raised by enhancing the clarity and coherence of the introduction, discussion, limitation, and future direction sections. Additionally, we have included a graphical abstract to provide a concise visual summary of our findings, which we believe will assist readers in grasping the key aspects of our research.

We understand your point regarding the diversity of study types and prostheses compared in our manuscript. To address this, we have taken extra care to effectively summarize and present the different studies, aiming to make the article more accessible and comprehensible to our readers. Please note the challenge of limited published literature on this particular topic, and we made the best possible selection of articles that closely align with the theme of personalization. Our intent was to provide a comprehensive overview despite the scarcity of directly relevant sources.

We sincerely hope that these revisions have made the manuscript more coherent and informative, aligning it better with your expectations and the standards of publication.

Reviewer 2 Report

The article is novel in that it addresses using customized imaging data to provide personalized prostheses, usually reserved for non cardiac procedures. Both congenital and anatomical heart disease could benefit from this approach. The conclusions reached affirm that this is a feasible approach using current and widespread imaging modalities. Methods and references seemed appropriate. But, there were no figures or tables.

Minor comments:

replace the word "heavy" medications line 50 with "using anticoagulation" or "medications with significant side effects"

take away "but" in line 159

in silico 2 words?

Author Response

We greatly appreciate your considerate feedback and the invaluable period of time you so kindly spent reading through our work. We truly appreciate your input, which is crucial in assisting us in enhancing and refining our research. We have included a point-by-point response to the comments and questions of the reviewers in the following section:

Reviewer -2

The article is novel in that it addresses using customized imaging data to provide personalized prostheses, usually reserved for non cardiac procedures. Both congenital and anatomical heart disease could benefit from this approach. The conclusions reached affirm that this is a feasible approach using current and widespread imaging modalities. Methods and references seemed appropriate. But, there were no figures or tables.

Response:

We are pleased to hear that you found our article to be novel and recognized its potential significance in applying customized imaging data for personalized prostheses in cardiac procedures. Indeed, we believe that both congenital and anatomical heart diseases can greatly benefit from this approach, and your acknowledgment of its feasibility using current imaging modalities aligns with our research objectives. In response to your feedback, we have incorporated one figure, one table, one supplementary table, and a graphical abstract into the manuscript. We believe these additions will enhance the clarity and visual representation of our findings, providing a more comprehensive and reader-friendly experience.

Minor comments:

replace the word "heavy" medications line 50 with "using anticoagulation" or "medications with significant side effects"

Response:

Thank you for your feedback. We have changed the sentence as per your suggestion.

take away "but" in line 159

Response:

Thank you for your feedback. We have removed “but” from that sentence.

in silico 2 words?

Response:

Thank you for your feedback. We have changed the “in silico” to two words.

Reviewer 3 Report

Dear Ladies and Gentlemans,

Unfortunately there is no clear scope within your manuscript as you mixed up several operations and interventions as well as human and animal studies. I guess if you try to focus on several topics, your article would be of interest for the society. 

can be improved

Author Response

Dear Reviewer,

We acknowledge your review and comments. We substantially revised the manuscript in response to the input. By improving the introduction, discussion, limitation, and future direction parts, we have allayed the issues brought up. In order to give readers a clear visual explanation of our findings, we have also included a graphical abstract. We hope that this will help readers understand the most important elements of our research.

Regarding the variety of study types and prostheses compared in our publication, we recognize your point. In order to address this, we took extra care to adequately summarize and display the many studies, with the goal of improving the article's accessibility and readership. Please be aware that there is a lack of published research on this specific topic.

Our aim was the find the potential importance of utilizing customized imaging data for tailoring prosthetic solutions in cardiac procedures. We firmly believe that this approach holds substantial promise for improving outcomes in patients with congenital and anatomical heart diseases. 

Reviewer 4 Report

Dear Author(s),

Subject: Feedback on the Manuscript "Personalized Surgery in Cardiovascular Replacement Therapy: A Systematic Review"

I am writing to communicate my feedback on your manuscript titled "Personalized Surgery in Cardiovascular Replacement Therapy: A Systematic Review". Firstly, I would like to commend you and your team for the exhaustive work and the in-depth analysis presented. The rigorous methodology and the comprehensive literature search are certainly praiseworthy.

However, to enhance the overall impact and clarity of the manuscript, I suggest considering the following points for improvement:

Expanded Introduction: While the current introduction provides a snapshot of the topic, a slightly expanded overview detailing the evolution of personalized surgery might be beneficial for readers unfamiliar with the subject.

Data Visualization: Although your tables and figures are informative, integrating a few more visual representations, perhaps a flowchart illustrating the systematic review process or a graphical abstract summarizing the main findings, could make the data more accessible.

Discussion on Limitations: All systematic reviews have inherent limitations. While you have touched upon some, a more explicit and dedicated subsection discussing these limitations would add depth to your manuscript and reflect transparency.

Future Directions: Given the rapidly evolving nature of the field, it would be beneficial for the readers if you could outline more specific future directions, potential challenges, and areas ripe for investigation. This will not only guide researchers but also stimulate thought and discussions among peers.

In conclusion, your manuscript is a solid piece of scientific literature that offers a significant contribution to the field of cardiovascular medicine. The aforementioned points are merely suggestions to further elevate its quality and resonance with the readership. I am optimistic about the positive impact your work will have in the scientific community.

Looking forward to seeing the finalized version of your work in JPM. If you have any questions or require further clarification on my feedback, please do not hesitate to reach out.

Kind regards,

CJR

Author Response

We greatly appreciate your considerate feedback and the invaluable period of time you so kindly spent reading through our work. We truly appreciate your input, which is crucial in assisting us in enhancing and refining our research. We have included a point-by-point response to the comments and questions of the reviewers in the following section:

Reviewer -3

Dear Author(s),

Subject: Feedback on the Manuscript "Personalized Surgery in Cardiovascular Replacement Therapy: A Systematic Review"

I am writing to communicate my feedback on your manuscript titled "Personalized Surgery in Cardiovascular Replacement Therapy: A Systematic Review". Firstly, I would like to commend you and your team for the exhaustive work and the in-depth analysis presented. The rigorous methodology and the comprehensive literature search are certainly praiseworthy.

Response:

We sincerely appreciate your kind words and the time you have dedicated to reviewing our manuscript. Your positive feedback is truly encouraging, and we are grateful for your thoughtful comments. We are pleased to hear that you found our work to be exhaustive and that you commend our team for the rigorous methodology and comprehensive literature search. We put significant effort into conducting a thorough analysis and ensuring the highest quality of research, and it is rewarding to receive recognition for these endeavors.

However, to enhance the overall impact and clarity of the manuscript, I suggest considering the following points for improvement:

Expanded Introduction: While the current introduction provides a snapshot of the topic, a slightly expanded overview detailing the evolution of personalized surgery might be beneficial for readers unfamiliar with the subject.

Response:

Thank you for your constructive feedback on our manuscript. In response to your recommendation, we have meticulously reviewed and re-written the introduction section of the manuscript. We have incorporated a more comprehensive overview, highlighted in yellow in the revised manuscript, which delves into the evolution of personalized surgery. This expanded introduction aims to provide readers, especially those less familiar with the subject, with a clearer understanding of the context and historical development of personalized surgery. We believe that these modifications will significantly improve the readability and accessibility of our manuscript, ensuring that it aligns more closely with the expectations of our audience.

Data Visualization: Although your tables and figures are informative, integrating a few more visual representations, perhaps a flowchart illustrating the systematic review process or a graphical abstract summarizing the main findings, could make the data more accessible.

Response:

In response to your recommendation, we have made several additions to enhance the visual representation of our findings. Specifically, we have included a flowchart as Figure 1 to illustrate the systematic review process, providing readers with a clear visual overview of our literature extraction method. A summary table has been incorporated to succinctly present key data and findings from our research. Additionally, we have included a supplementary table to provide supplementary information that complements our main findings.

As per your suggestion, we have included a graphical abstract to summarize the main findings of our study, making it more accessible to our readers.

Discussion on Limitations: All systematic reviews have inherent limitations. While you have touched upon some, a more explicit and dedicated subsection discussing these limitations would add depth to your manuscript and reflect transparency.

Response:

Thank you for your thoughtful feedback. we have re-written the limitation section of our manuscript to create a more explicit and dedicated subsection. This revised section aims to provide a comprehensive discussion of the inherent limitations associated with our systematic review. We believe that this addition will not only add depth to our manuscript but also enhance transparency in acknowledging and addressing potential constraints.

Future Directions: Given the rapidly evolving nature of the field, it would be beneficial for the readers if you could outline more specific future directions, potential challenges, and areas ripe for investigation. This will not only guide researchers but also stimulate thought and discussions among peers.

Response:

We greatly appreciate your feedback. In accordance with your recommendation, we have separated the section on future areas of work from the limitation section to create a dedicated subsection titled "Future Directions."

In this new subsection, "Future Directions," we have outlined specific future directions, potential challenges, and areas that are ripe for investigation. Our goal is to provide readers with a more detailed roadmap for potential research avenues in this rapidly evolving field. We believe this addition will not only guide fellow researchers but also stimulate discussions and further exploration among peers.

In conclusion, your manuscript is a solid piece of scientific literature that offers a significant contribution to the field of cardiovascular medicine. The aforementioned points are merely suggestions to further elevate its quality and resonance with the readership. I am optimistic about the positive impact your work will have in the scientific community.

Looking forward to seeing the finalized version of your work in JPM. If you have any questions or require further clarification on my feedback, please do not hesitate to reach out.

Kind regards,

CJR